# Flammability and Thermoregulation Properties of Knitted Fabrics as a Potential Candidate for Protective Undergarments

**DOI:** 10.3390/ma15072647

**Published:** 2022-04-04

**Authors:** Laimutė Stygienė, Sigitas Krauledas, Aušra Abraitienė, Sandra Varnaitė-Žuravliova, Kristina Dubinskaitė

**Affiliations:** 1Center for Physical Sciences and Technology, Department of Technological Development of Textile, Demokratų Str. 53, LT-48485 Kaunas, Lithuania; laimute.stygiene@ftmc.lt; 2Center for Physical Sciences and Technology, Department of Textile Technologies, Demokratų Str. 53, LT-48485 Kaunas, Lithuania; sigitas.krauledas@ftmc.lt (S.K.); ti@ftmc.lt (A.A.); 3Center for Physical Sciences and Technology, Department of Textiles Physical-Chemical Testing, Demokratų Str. 53, LT-48485 Kaunas, Lithuania; kristina.dubinskaite@ftmc.lt

**Keywords:** non-flammable knitted fabrics, protective underwear, thermoregulating properties

## Abstract

The most important functional purpose of knitted fabrics used for the protective non-flammable underwear worn in contact with the skin is to ensure wearing comfort by creating and maintaining a constant and pleasant microclimate at the skin surface independently from the environmental conditions. Protective non-flammable underwear may be used by firefighters or sportsmen, e.g., racing (Formula) sportsmen, where a risk of burn injuries (when the car is on fire after a car crash) is present. In order to investigate the flammability and thermal comfort properties of two-layer knitted fabrics, two groups of aramids and flame-retardant (FR) viscose fiber fabrics of different combined patterns and surface structures (porosity and flatness) were designed and manufactured for this research. Aramid fiber spun yarns (METAFINE.X.95^®^) formed the inner layer (contacting with human skin) of fabrics and aramid/viscose FR fiber spun yarns (METALEN^®^) formed the outer layer. For the evaluation of the functional characteristics of the manufactured fabrics, the flammability and thermoregulating properties, such as liquid moisture management, water vapor and air permeability, and thermal resistance were investigated. The results show that all tested fabrics are non-flammable, breathable, permeable to air, and can be assigned to moisture management fabrics. Their obtained overall moisture management capacity (OMMC) values are in the range 0.59–0.88. The knitted fabrics with an embossed porous surface to skin had a higher OMMC (0.75–0.88). The thermoregulation comfort properties were mostly influenced by the structure of the fabrics, while the burning behavior was found to be independent from the structure, and the non-flammability properties were imparted by the fiber content of the knits.

## 1. Introduction

Special and highly functional knitted fabrics are usually used for the manufacturing of non-flammable underwear, which are worn in touch with the skin by humans in extreme conditions, such as firefighters, soldiers, policemen, or even racing sportsmen. Such fabrics are currently made of flame retardant or non-flammable materials, whose efficiency depends on the limiting oxygen index (LOI) of the fibers used [1]. The LOI is the minimum percentage of oxygen it takes in an airlike gas mixture to support flaming combustion. The LOI of some popular non-flammable fibers are as follows: meta-aramid: 28.5–30; para-aramid: 29; and Lenzing FR (FR stands for flame resistant) fibers: 28 [2,3]. Apart from flammability, the thermophysiological comfort of the non-flammable underwear is one of the most significant factors. All these factors depend on the fiber content, fabric structure, and finishing procedures [4,5,6,7,8,9]. The thermophysiological comfort is dictated by the moisture management properties, thermal resistance, air permeability, and water vapor permeability of the fabrics. Although the human body is self-thermoregulating and tries to maintain a core body temperature of 37 °C, low water-vapor resistance and high air permeability of fabrics induce evaporation of perspiration and cooling of the body [1,4]. All these properties improve wearing comfort and reduce physiological stress. If the outer clothing cannot transport perspiration in vapor form away from the wearer’s skin sufficiently quickly, perspiration condenses on the skin (or the inner fabric surface) [1,4]. In that case, the underwear should transmit the moisture away from the skin or inner fabric surface to the outer surface as quickly as possible for the protection and comfort of the wearer [1,10]. That is why a fabric’s dynamic liquid and moisture transport properties are very important in order to reduce and control the humidity on the surface of the skin [11,12]. The multi-dimensional liquid diffusion capacity of porous fabrics is evaluated by investigating liquid and moisture transfer in all directions of the fabrics, which are dynamically coupled with the heat transfer process [13,14]. Synthetic underwear plays a significant role in moisture and metabolic heat transfer within firefighter clothing, and through total heat loss measurements [15,16], it is important to analyze the liquid moisture management properties of firefighter clothing with a new fire-resistant double-layered knitted fabric underwear [17,18]. The aim of the research was to design, develop, and produce two-layer knitted fabrics for use as underwear for firefighters, racing (Formula) sportsmen, and other professionals, which have very good thermoregulating comfort properties combined with non-flammability. As many non-flammable fabrics are uncomfortable to wear even as outer clothing, selected knitting patterns were used, which formed the unique structure of the designed knitted fabrics. Two types of aramid-based yarns were utilized in order to impart the textiles with non-flammable properties. For the investigation of the level of the fabrics’ comfort, moisture management properties, water vapor permeability, thermal resistance, air permeability, and fabrics’ structural parameters were determined and estimated.

## 2. Materials and Methods

Two groups of knitted fabrics were designed and manufactured from METAFINE.X.95^®^ and METALEN^®^ (linear density of each yarn: 16.4 tex) spun yarns for investigations. Each group consisted of eight double-layered weft-knitted fabrics with different combined pattern. Fabrics were knitted on a circular 22E gauge rib knitting machine. All produced untreated knitted fabrics were washed with non-ionic detergent (2 g/L of Felosan RG-N) and 0.5 g/L of sodium carbonate at 60 °C for 30 min in the washing machine (WASCATOR FOM71MP-Lab), rinsed, centrifuged, and then dried in the laboratory oven (TFO/S IM 350) at 100 °C. METAFINE.X.95^®^ by Filidea S.r.l. (Italy) is a registered trademark of spun yarns with inherently flame-resistant fibers of the aromatic polyamide (aramid) family. The fiber composition of yarns is as follows: 95% meta-aramid (linear density: 1.7 dtex; fiber length: 51 mm) and 5% para-aramid (linear density: 2.2 dtex; fiber length: 51 mm). METALEN^®^ by Filidea S.r.l. (Italy) is a registered trademark of spun yarns, manufactured from 50% meta-aramid and 50% hygroscopic viscose Lenzing FR^®^ modified fiber (linear density of each: 2.2 dtex; fiber length: 51 mm) (microscopic images of fibers are presented Figure 1). The thermal profile of the yarns cannot be provided because aramid fibers have no melting point and the other transition temperatures are very high. Lenzing FR^®^ fibers were produced by Lenzing AG (Austria) with phosphorus/sulphur containing additives. Phosphorous-containing FR agents incorporated in the fibers act according to the following principle: the production of polyphosphoric acid and carbonization takes place during burning and a protective layer is formed on the surface of a fabric. These phosphorous-containing FR agents decompose before the cellulose decomposes. The application of hygroscopic FR fibers in the protective clothing is crucial for the comfort of the protective textiles. The combined pattern was chosen for the designed and manufactured fabrics in which number of stitches (courses and wales) per centimeter was 16 and 11, respectively. The aramid yarns formed the inner layer and meta-aramid/viscose yarns formed the outer layer of the fabric. The knitted structures of samples 1A–4A and 1B–4B were developed so that the inner side (next to the skin) of the knitted fabric was imparted with an embossed porous surface. The outer surface of samples 5A–8A and 5B–8B was flat. The patterns of all the developed knitted structures are presented in Table 1 (additionally see in footnotes) and their main characteristics are shown in Table 2.

The number of stitches was calculated according to LST EN 14971 [19] standard, and the mass per unit area was determined according to LST EN 12127 [20] standard. The mean loop length *l* of investigated knitted fabrics was determined from the theoretical area density expression [21]:(1)l=A·B·M/T;
where *l* is the mean loop length of the knitted fabric in mm; *M* is the mass per unit area of the knitted fabric in g/m^2^; *A* is the wale spacing of the knitted fabric in mm; *B* is the course spacing of the knitted fabric in mm; *T* is the linear density of the yarns in tex.

For determination of the tightness factor *TF*, the following formula [22,23] was used:(2)$$TF=\sqrt{T}/l;$$

Porosity *E* and the volume filling rate *E_v_* were calculated using equations [24,25,26]:(3)$$E=l-E_{v};$$
(4)$$E_{v}=\pi \cdot d^{2}\cdot l/4A\cdot B\cdot h;$$
where *l* is the mean theoretical loop length of the knitted fabric in mm; *A* is the wale spacing of the knitted fabric in mm; *B* is the course spacing of the knitted fabric in mm; *d* is conventional diameter of the yarns in mm; *h* is the thickness of the knitted fabrics in mm.

The thickness of the knitted fabrics was determined with a DM-teks thickness apparatus (under an applied load of 50 g/cm^2^) according to method LST EN ISO 5084 [27].

The conventional diameter *d* (in mm) of the yarns (see Table 3) was calculated according to the following equation:(5)$$d=2/r\pi^{\frac{1}{2}}\cdot {\left(T/\rho \right)}^{1/2};$$
where *T* is the linear density of yarns in tex and *ρ* is fiber density in kg/m^3^.

To determine the multi-directional liquid moisture transport capabilities of the knitted fabrics a moisture management test (MMT) device, model M290, SDL Atlas (a view of the sensors is presented in Figure 2) was used. The investigations were carried out according to the American Association of Textile Chemists and Colourists (AATCC) Test Method 195 [28]. Five specimens with dimensions of 8 × 8 cm were prepared for each type of fabric. The specimens were then conditioned in a conditioning atmosphere (temperature (20 ± 2) °C, relative humidity (65 ± 4)%) and tests were carried out in the same conditioning atmosphere. MMT properties of knitted fabrics were evaluated by placing a fabric specimen between two horizontal (upper and lower) electrical sensors. The top surface (facing the top sensor) during testing was the inner side (next to the skin) of the fabric. A predefined amount of test solution (synthetic sweat) was introduced onto the inner side of the fabric. The test solution was free to move in three directions after being dropped onto the fabric’s surface: spreading outward on the surface of the fabric, transferring through the fabric from the inner side to the face side of the fabric, and spreading outward on the face side of the fabric and then evaporating. A summary of the measurement results was used to grade the liquid moisture management properties of the fabric using predetermined indices. MMT indices used in the investigation and their grading according to AATCC Test Method 195 are presented in Table 4.

In order to ensure the thermophysiological comfort under different environmental conditions or at various levels of activity, various factors other than the moisture management properties, i.e., moisture (liquid sweat) transfer from the body to the outside, are important. For example, the water vapor permeability (“breathing”) and the thermal resistance can play a key role in the thermophysiological comfort. According to standard CEN/TR 16422 [29], depending on the environmental conditions and the level of physical activity, the thermoregulating properties for skin contact materials are classified into three performance levels: A (very good), B (good), and C (medium). The main characteristics of thermoregulatory properties according to these levels for materials in direct contact with human skin are given in Table 5.

The air permeability was determined according to LST EN ISO 9237 [30] standard. The air permeability was measured using the differential pressure air permeability tester FAP-1034-LP (Frazier Precision Instrument Company Inc. Tm., (USA)), at a pressure drop of 100 Pa. The test surface area of 20 cm^2^ was used and 10 places for each fabric were measured in order to improve the accuracy. The coefficient of variation of measurements was approximately 7%. The conditioning of samples and testing were carried out in standard atmospheres.

The water vapor permeability was determined using the modified “cup” method. This is measured as the amount of water that evaporates from the covered vessel in 24 h. Equipment used was as follows: thermostat U10 with the water bath maintained at 38 °C; a thermostatic receptacle made of steel with an inside diameter of 88.2 mm; a 500 mL measuring container; and weighing scales, with an accuracy of 0.01 g. The specimens were prepared as follows: three 115 mm diameter specimens were cut from different parts of the material, weighed, and maintained in a standard climate for 24 h. Then, the dish containing 500 mL of distilled water was placed in the thermostat. The sample was firmly secured to the pan for 6 h. The ambient temperature during the tests was −22.0 ± 0.1 °C. Investigation stand is shown in Figure 3. The arithmetic mean of five specimens for each fabric was counted and the coefficient of variation was not more than 5%.

Thermal resistance was measured using the sweating guarded-hot plate method according to LST EN ISO 11092 [31] standard, where the temperature of a hot plate was 35 °C, and the air temperature in the camera was 20 °C. The apparatus used was the Sweating Guarded Hot Plate M259B, produced by SDL International Ltd. (England). The mean out of 3 specimens for each fabric was taken for the determination of thermal resistance. The coefficient of variation was not more than 5%. As the mean values of air permeability, water vapor permeability, and thermal resistance of each group of fabrics (groups: 1A–4A, 5A–8A, 1B–4B, 5B–8B) were very similar, the mean values and standard deviation of each group’s results were calculated and presented as a mean value of a parameter.

The burning behavior of the tested fabrics was assessed according to LST EN ISO 15025 [32] standard (procedure A: surface ignition) with the Rhoburn model 480 (England) flammability tester using propane gas. The flame application time was 10 s, the flame height was (25 ± 2) mm, and the ambient conditions in the test camera were a temperature of 19 °C and a relative humidity of 45%. Three specimens from each fabric in the longitudinal and cross directions were cut for this test and the mean values and observations were presented as a result. The calculations of the charred area and its perimeter in the flame application zone were performed using the ImageJ software. The coefficient of variations of the calculations obtained using the ImageJ software was approximately 2%.

## 3. Results

Thermal comfort and moisture management properties of fabrics are very important for underwear. Such garments have to be comfortable, breathable, able to transfer moisture from the body to the outside, while still providing a warm and comfy feeling to the wearer. All parameters indicating moisture management properties are summarized in Table 6, where it can be seen that all the knitted fabrics may be assigned the title “moisture management fabrics”.

All the tested fabrics can be also characterized as having medium to fast wetting, medium to fast absorption, a large spread area on the bottom surface, fast spreading on the bottom surface, and good-to-excellent one-way transport (see Table 4). The tortuosity (τl) of moisture expresses the tortuous path of the liquid flow through textiles. However, it is generally known that the tortuous path in a transported material, on the micro-scale, is very complicated. Analytically the tortuosity of a liquid is defined as the ratio of the actual length of the straight length or thickness of a sample to the actual length of the flow path [13,22]. WTB better reflects the dynamic dimensional liquid transfer capability. Considering the difficulty to measure the actual flow path through textiles, the effective value of tortuosity of a liquid may be assumed to be equal to the ratio of the wetted time of the top surface to that of the top and bottom surface (WTT and WTB) (see Table 6).
τl = WTT/WTB(6)

When WTB = 0, it means that there is no liquid transferred across textiles, and the tortuosity of liquid through textiles is supposed to be infinitely large [13]. The data in Table 6 and Figure 4 show that a larger WTB results in more liquid transferred to the bottom surface and a smaller tortuosity of liquid through textiles. The same conclusion was reported by Aihua and Yi in [13]. It can be also seen from results presented in Figure 3 that the tortuosity values of fabrics 5A–8A and 5B–8B are very similar, i.e., approximately 1.2. This is because the pattern of the outer layer of these fabrics has a honeycomb structure, the porosity of the fabrics is less, and the tightness factor and volume filling are higher as compared with the other tested fabrics (see Table 2). In [13], it is also stated that, with the increase in WTB, the water vapor concentration in textiles becomes higher. This is because the higher the WTB, the longer the liquid transfer tortuosity, and thus less water vapor is transferred out and evaporated.

The ability of the tested fabrics to absorb liquid is demonstrated in Figure 5. It can be seen from Figure 5b that the absorption rate of the outer surface of all the tested groups (A and B) is very similar due to the honeycomb structure of the fabrics (see Table 1 and Table 2), while the absorption rate of the inner surface obviously differs among fabrics and fabric surfaces. This may be because moisture diffusion into a fabric through air gaps between yarns and fibers is a fast process, while moisture diffusion into fibers is coupled with the heat-transfer process, which is much slower and is dependent on the ability of fibers to absorb moisture [33,34,35].

The parameter of the maximum wetted radius (MWR) is very important for human comfort. The textile with the smaller MWRB has less area for effective evaporation. This means that when the human body starts to sweat, more perspiration is accumulated in the textiles. This may cause a wet sensation and an uncomfortable feeling to the human body. As the MWRB of all tested fabrics is more than 20 mm (see Figure 6b), which is assigned to grades 4 and 5, the person wearing the underwear made from the tested fabric will not feel uncomfortable. The MWRT parameter is not very important for the evaluation of human sensations.

Figure 6a shows the relationship between the liquid spreading speed of the top (SST) side of both groups of tested fabrics (groups A and B) and TF. The SST was defined as the accumulated rate of surface wetting from the center of the specimen where the test solution was dropped to the maximum wetted radius. As in [33], only for SSB, the spreading speed of the top surface (SST) of the fabrics decreased with the increase in the tightness factor of the tested fabrics. A high determination coefficient improves the relationship between these two parameters, A decrease in the spreading speed is caused by a decrease in porosity (see Table 2). Moreover, the solution spreads faster on the fabrics that have the highest rates of surface energy, i.e., knitted fabrics from viscose. The liquid spreads from very fast to fast on the bottom surface of group A and B fabrics, as is illustrated with data in Figure 7a.

Figure 8 shows the mean grade of the cumulative moisture difference between the two surfaces of the fabric—the accumulative one-way transport index (AOTI). It can be seen from the presented data that the AOTI is the best (excellent) for fabrics 1B–4B, while for other fabrics, it can be categorized as very good or even good. The same tendency is observed for the overall moisture management capacity (OMMC) of the fabrics tested (see Figure 9). Therefore, it can be stated that, despite all the tested fabrics falling into the category “moisture management fabrics”, the best moisture management properties were found for fabrics 1B–4B. Furthermore, it was noted that increasing the tightness factor of the knitted fabrics resulted in a decreased overall moisture management capacity (OMMC) (see Table 2 and Figure 9).

The correlation between the two most important moisture management indicators of all knitted fabrics investigated, i.e., AOTI and OMMC, is presented in Figure 10. It can be seen that a strong linear relationship exists between these parameters (R^2^ = 0.86). The strongest correlation between these parameters can be observed for fabrics 1A–8A and 5B–8B (R^2^ = 0.97), for which almost all moisture management parameters were very similar. Such results were influenced by the structure of the knitted fabrics, which is formed by the knitting pattern. The same conclusion was reported by the authors in [4,33,35].

It is known that textiles with good liquid moisture management properties are usually composed of at least two layers. The main requirement for the inner layer in contact with human skin is to absorb and transfer moisture effectively to the outside. The outer layer of the material must effectively distribute moisture so that it evaporates rapidly into the environment. Several principles of production of double-layer sweaters with good moisture transport can be distinguished. The first is when hydrophobic synthetic fibers of ordinary or unique structures are used to form the inner layer of the knit, and hydrophilic fibers are used to form the outer layer. According to the second principle, double-layer knits are formed in both layers using only hydrophobic synthetic fibers with different geometric parameters. According to the third principle, the hydrophilicity of the outer layer must be higher than that of the inner layer. The need for a two-layer knitted fabric for the design and manufacture of protective underwear can be explained by the fact that the two-layer structure provides good liquid moisture management properties. In our case, the moisture regain of the fibers in the inner layer is about 5.5. The outer layer is formed from a mixture of fibers with higher hydrophilicity, since the moisture regain of the existing 50% viscose FR fiber is 7.11% [36]. This improves the transfer of sweat liquid from the inner layer to the outer. The good moisture management properties of the knits were clearly confirmed by the OMMC results obtained.

In order to investigate and determine all the parameters that influence thermoregulation comfort for humans, water vapor permeability, thermal resistance, and air permeability tests were performed on the tested fabrics. As one group consisted of four fabrics and all data received during tests were very similar among the group’s fabrics, the mean values for each group out of four was calculated, as shown in Table 7. The thermal resistance of the tested fabric groups varied from 0.052 to 0.082 m^2^ K/W. Regarding standard CEN/TR 16,422, all the tested fabrics are intended to be worn next to the human body in cold climate conditions (see Table 5). The knitted fabrics from group 1B ÷ 4B with thermal resistance values of Rct ≥ 0.08 m2 K/W, which were characterized as having the best moisture management properties (OMMC = 0.86–0.88) according to CEN/TR 16,422, correspond to the highest (A) performance level intended for skin contact materials. Similar results for non-flammable viscose blended knitted fabrics were reported by Glombikova and Komarkova in [1]. The results of the water vapor permeability parameter demonstrate that the manufactured fabrics are breathable and may be used as underwear fabrics. High air permeability values confirm the comfort of materials as air is not captured inside.

The data presented in Figure 11 shows that there is a strong linear relationship between air permeability and the tightness factor of fabrics (R^2^ = 0.9). Higher values of TF correspond to higher values of air permeability.

All results presented in Table 7 support the main hypothesis that the manufactured tested fabrics can be used as underwear materials.

The burning tests demonstrated that the fabrics may be used for the production of non-flammable underwear, because they do not ignite after 10 s (a requirement according to EN ISO 15025). The evaluation of flame resistance was performed using image analysis in order to detect differences in non-flammability in the fabrics of different structures, which exhibit the same levels of flame resistance. The results of the charred area parameters in the fabrics after flame application, as presented in Table 8, show that, despite the different structures of the tested fabrics, the area, perimeter, length, and width of the burn territory were very similar in all materials (for more information see Figure 12). This may be because the fiber content and yarns used for manufacturing the knits were the same, and only this influenced the flame path, because no other correlation was detected.

The authors in [1] also investigated non-flammable materials intended for underwear. They reported very similar burning test results for viscose blended fabrics as those in this paper.

## 4. Conclusions

Textiles with good liquid moisture management properties are usually composed of at least two layers. The main requirement for the inner layer, which is in contact with human skin, is to absorb and transfer moisture effectively to the outside. The outer layer of material must effectively distribute moisture so that it evaporates rapidly into the environment. Our investigations into manufactured non-flammable knitted fabrics showed that, for knitted fabrics with an embossed porous surface to the skin, the best OMMC of 0.86–0.88 was observed for knitted fabrics 1B÷4B. Their OMMC rates are excellent, with an OMMC index of 5.

This shows that the OMMC parameter depends on the knitting pattern of the fabrics, i.e., better results were observed when the press loops in the knit structure were formed in every second column.

Comparing the knits of group A (press loops were formed every fourth column in the fabric structure), a marginally better OMMC (0.75–0.80) was observed for fabrics 1A–4A, where the porous surface was used next to the skin while wearing; however, there is no essential difference, because the OMMC rates of knits1A–4A and 5A–8A are good/very good.

Comparing the knits of group B (press loops were formed every second column in the fabric structure), a significantly better OMMC 0.86–0.88 was observed for fabrics 1B–4B, where the porous surface was next to the skin while wearing. The OMMC rates of these fabrics were excellent, while the OMMC of fabrics 5B–8B was (0.59–0.72), which is very good.

The investigations demonstrated that, regarding standard CEN/TR 16,422, all tested fabrics can be worn next to the human body in cold climate conditions. The knitted fabrics of group 1B–4B with thermal resistance values of Rct ≥ 0.08 m^2^ K/W, which were characterized as having the best moisture management properties (OMMC = 0.86–0.88), according to CEN/TR 16422, corresponds to the highest (A) performance level intended for skin contact materials. The water vapor permeability and air permeability parameters demonstrate that the manufactured fabrics are breathable, permeable to air, and may be used as underwear fabrics. The evaluation of flame resistance of the fabrics showed that the fabrics are non-flammable, but there no significant differences were seen in the non-flammability behavior of the fabrics. This led to the conclusion that the fiber composition, which was the same for each material, was the main factor influencing these results. All the test results support the main hypothesis that the manufactured tested fabrics can be used as underwear materials.
**No. of Fabric****Inner Layer (in Contact with the Skin)****Outer Layer**1A
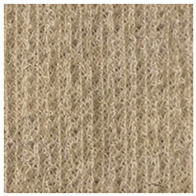

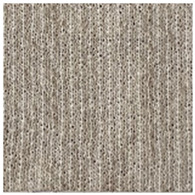
2A
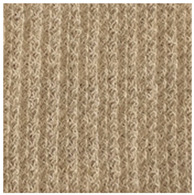

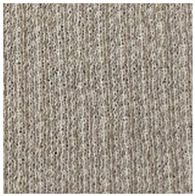
3A
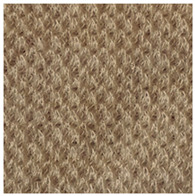

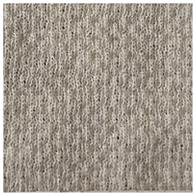
4A
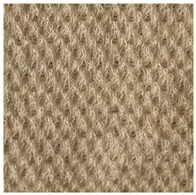

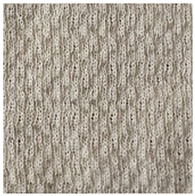
5A
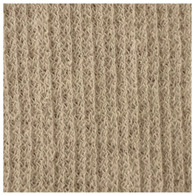

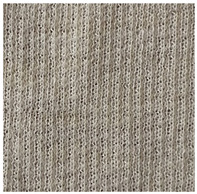
6A
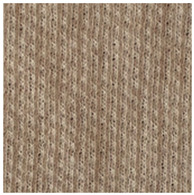

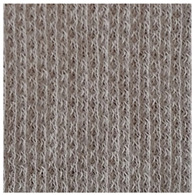
7A
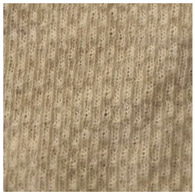

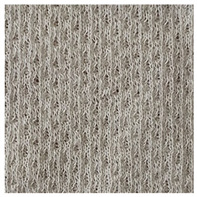
8A
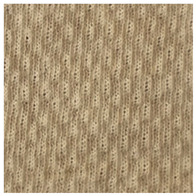

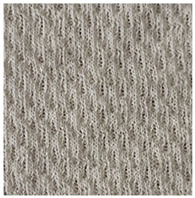
1B
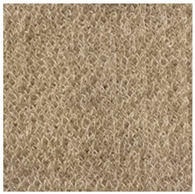

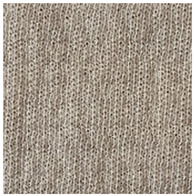
2B
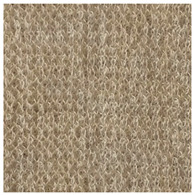

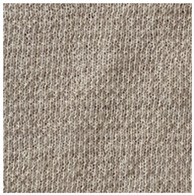
3B
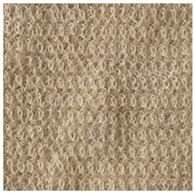

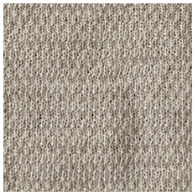
4B
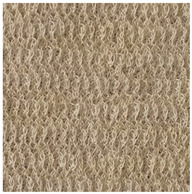

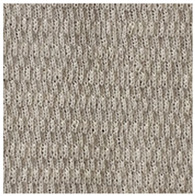
5B
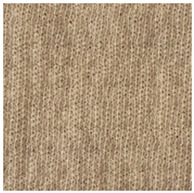

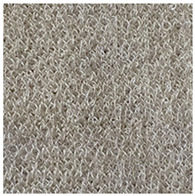
6B
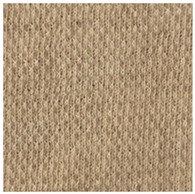

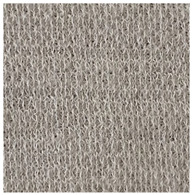
7B
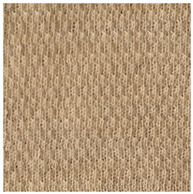

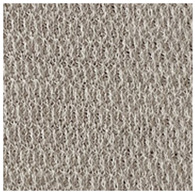
8B
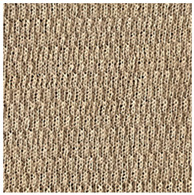

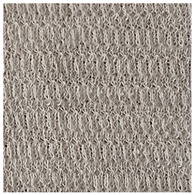


## Figures and Tables

**Figure 1 materials-15-02647-f001:**
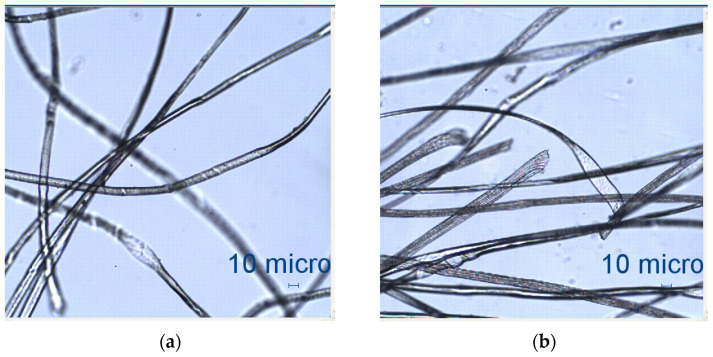
Microscopical view of meta-aramid and para-armid fibers (METAFINE.X.95^®^ yarns) and meta-aramid and viscose FR fibers (METALEN^®^ yarns) (magnification 10×). (**a**) Meta-aramid and para-armid fibers (METAFINE.X.95^®^ yarns). (**b**) Meta-aramid and viscose FR fibers (METAFINE.X.95^®^ yarns).

**Figure 2 materials-15-02647-f002:**
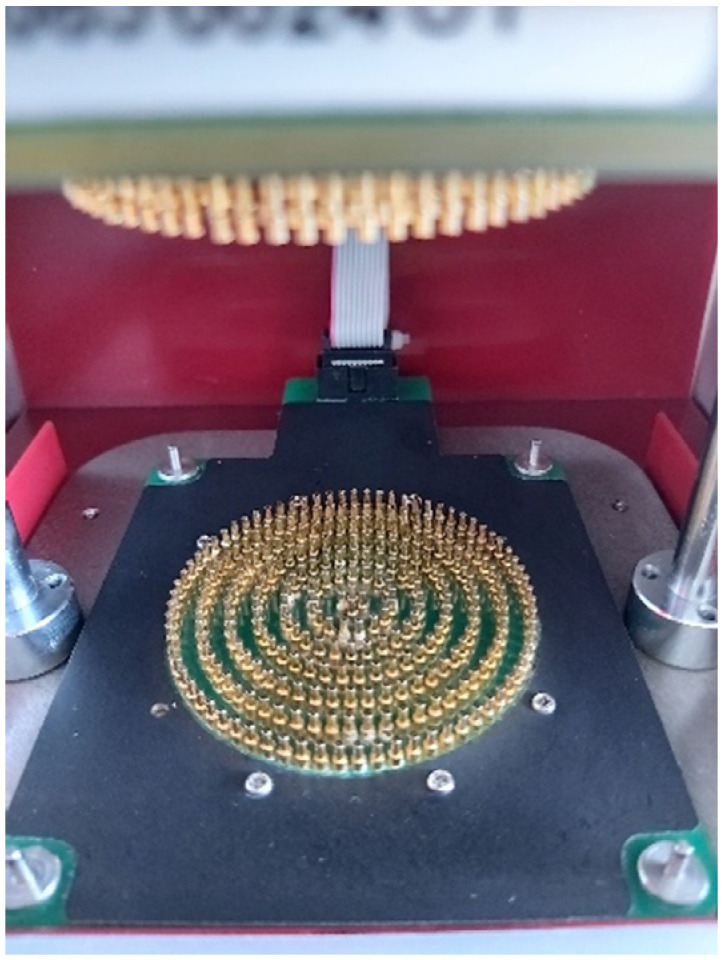
A view of the tester sensors MMT290.

**Figure 3 materials-15-02647-f003:**
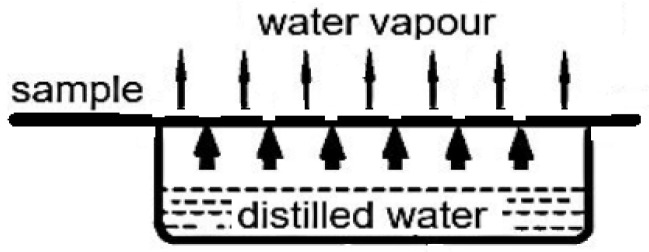
The laboratory setup for water vapor permeability (the cup method).

**Figure 4 materials-15-02647-f004:**
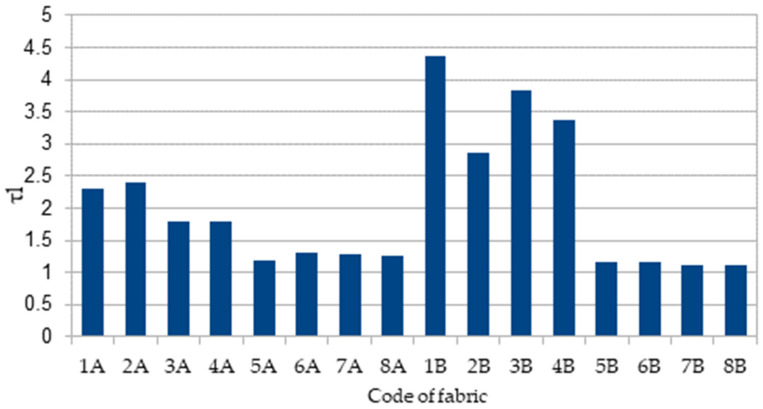
The tortuosity of liquid in the tested knitted fabrics.

**Figure 5 materials-15-02647-f005:**
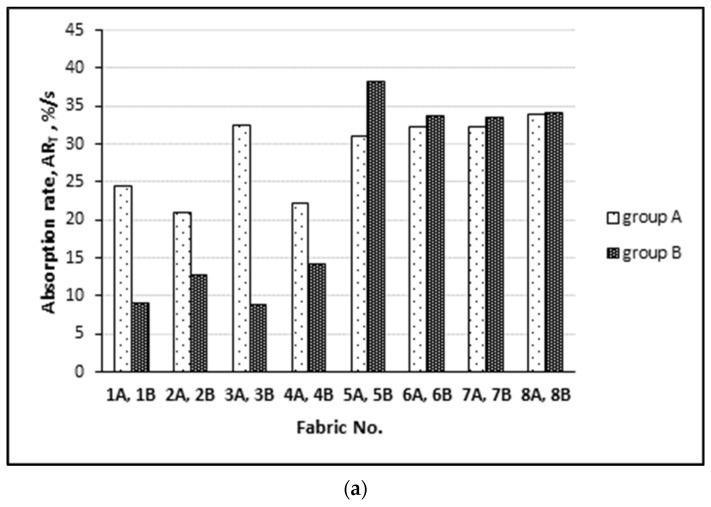
Absorption rate of the knitted fabrics investigated on: (**a**) the inner surface of the fabrics; (**b**) the outer surface of the fabrics.

**Figure 6 materials-15-02647-f006:**
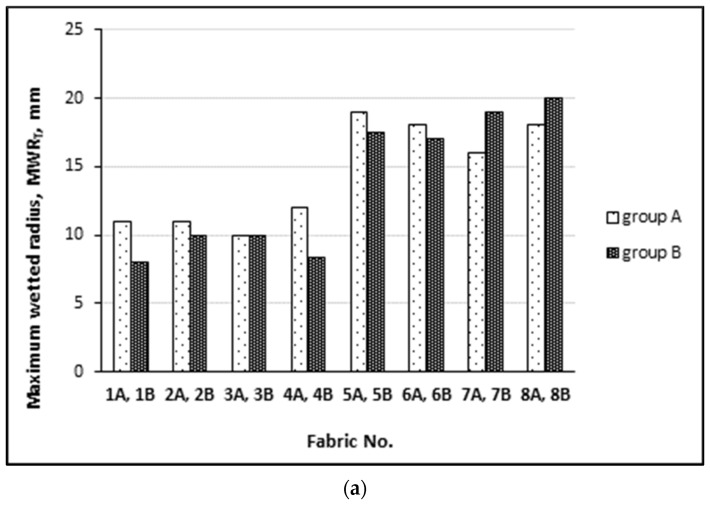
Maximum wetted radius on: (**a**) the inner surface of the fabrics; (**b**) the outer surface of the fabrics.

**Figure 7 materials-15-02647-f007:**
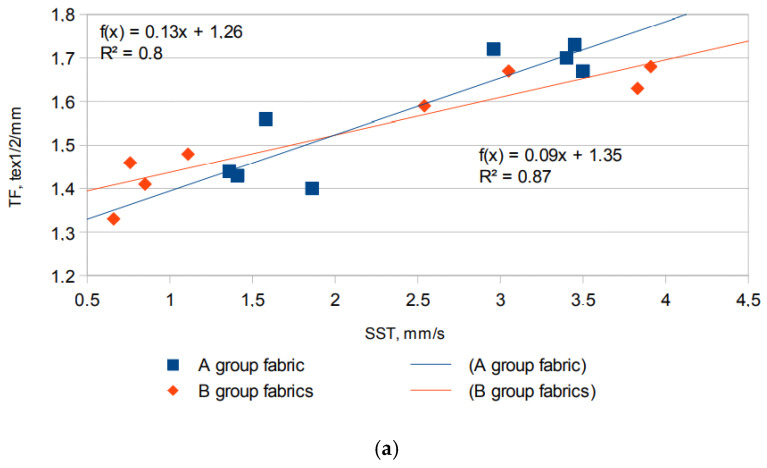
Graphical view of: (**a**) relationship between the spread speed of the top (inner) surface and the tightness factor; (**b**) spreading speed on outer surface of fabrics.

**Figure 8 materials-15-02647-f008:**
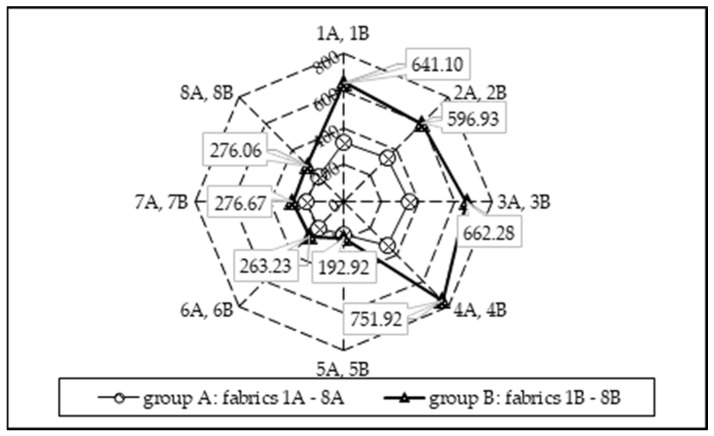
Accumulative one-way transport capability (AOTI) of the fabrics tested.

**Figure 9 materials-15-02647-f009:**
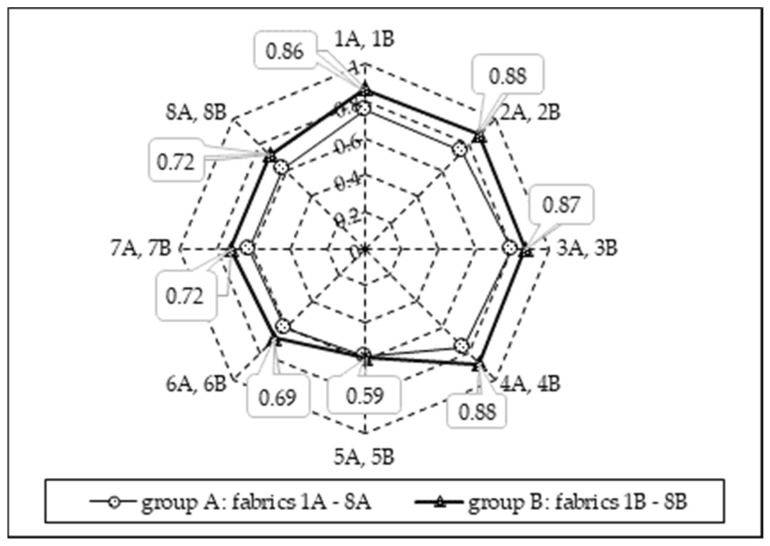
Overall moisture management capacity (OMMC) of the fabrics tested.

**Figure 10 materials-15-02647-f010:**
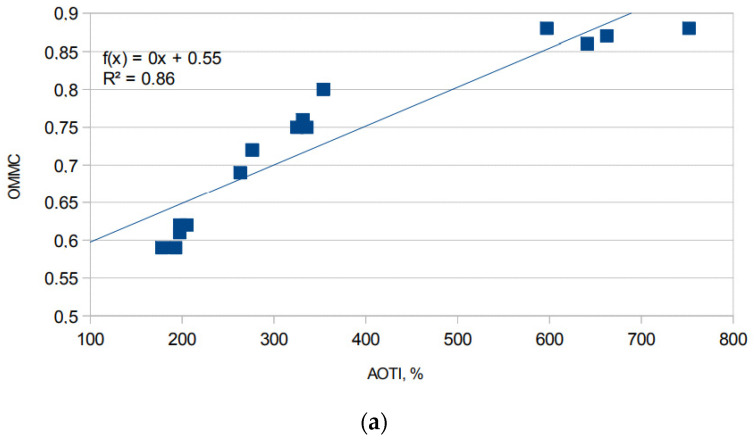
Relationship between AOTI and OMMC: (**a**) for all the tested fabrics 1A–8A and 1B–8B; (**b**) for the tested fabrics 1A–8A and 5B–8B.

**Figure 11 materials-15-02647-f011:**
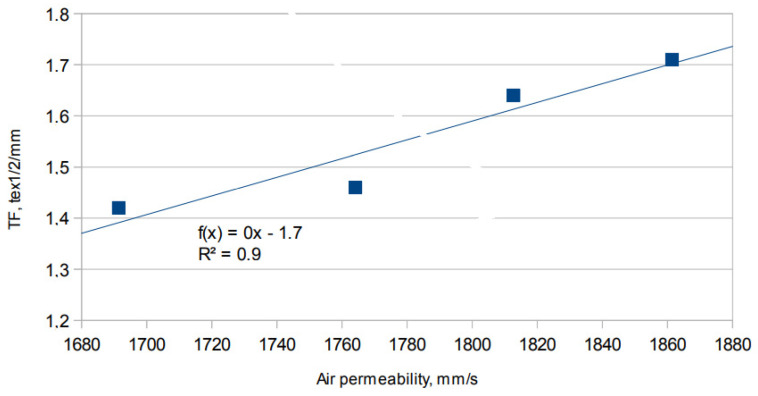
Relationship between air permeability and the tightness factor of the tested fabrics.

**Figure 12 materials-15-02647-f012:**
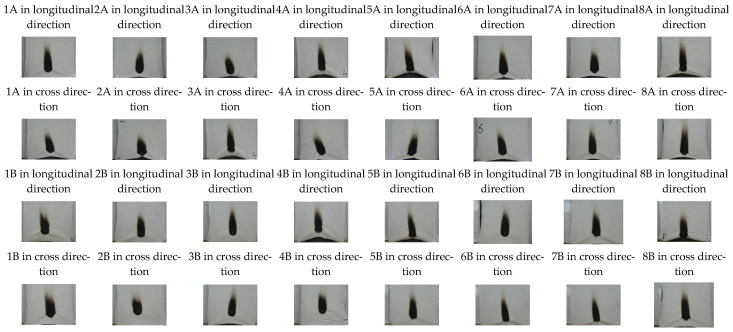
Charred area images after flame application to the fabrics.

**Table 1 materials-15-02647-t001:** The main characteristics of two-ply twisted yarns with new structures.

Code of Pattern	Pattern of Knitted Fabric	Code of Pattern	Pattern of Knitted Fabric
I A		I B	
II A	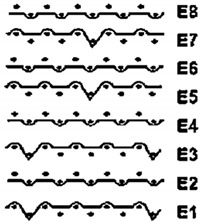	II B	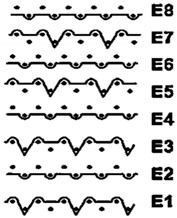
III A	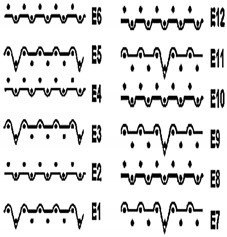	III B	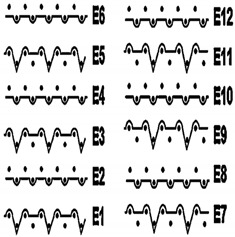
IV A	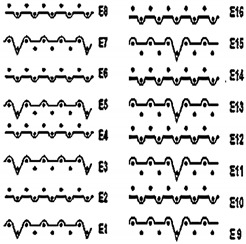	IV B	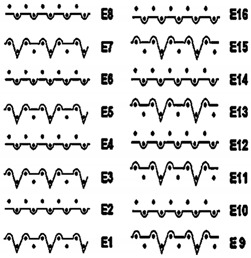

**Table 2 materials-15-02647-t002:** The main structural characteristics of two-layer weft knitted fabrics.

No. of Fabric	Pattern Code (See Table 1)	Type of Separate Layer *	Type of Flame-Retardant Spun Yarn, Linear Density (Tex)	Pattern Courses (See Table 1)	Mass Per Unit Area (g/m^2^)	Mean Loop Length L, mm	Tightness Factor TF (tex^1/2^/mm)	Thick-Ness h (mm)	Volume Filling Rate E_v_	Porosity E, %
1A	IA	Inner	METAFINE.X.95^®^, 16.4	E1, E3…–E15 (uneven)	212	3.67	1.56	1.27	0.23	77.37
2A	IIA				230	3.98	1.44	1.37	0.23	77.24
3A	IIIA		METALEN^®^, 16.4	E2, E4…–E16 (even)	231	4.00	1.43	1.40	0.22	77.63
4A	IVA	Outer			236	4.09	1.40	1.43	0.22	77.63
5A	IA	Inner	METAFINE.X.95^®^, 16.4	E2, E4…–E16 (even)	191	3.31	1.73	0.97	0.27	73.31
6A	IIA				195	3.38	1.70	1.02	0.26	74.08
7A	IIIA		METALEN^®^, 16.4	E1, E3…–E15 (uneven)	192	3.33	1.72	1.05	0.25	75.21
8A	IVA	Outer			198	3.43	1.67	1.10	0.24	75.60
1B	IB	Inner	METAFINE.X.95^®^, 16.4	E1, E3…–E15 (uneven)	248	4.30	1.33	1.40	0.24	75.99
2B	IIB				224	3.88	1.48	1.37	0.22	77.83
3B	IIIB		METALEN^®^, 16.4	E2, E4…–E16 (even)	234	4.05	1.41	1.33	0.24	76.15
4B	IVB	Outer			226	3.92	1.46	1.32	0.23	76.79
5B	IB	Inner	METAFINE.X.95^®^, 16.4	E2, E4…–E16 (even)	208	3.60	1.59	1.11	0.25	74.60
6B	IIB				198	3.43	1.67	1.07	0.25	74.91
7B	IIIB		METALEN^®^, 16.4	E1, E3…–E15 (uneven)	197	3.41	1.68	1.05	0.25	74.57
8B	IVB	Outer			203	3.52	1.63	1.03	0.27	73.28

* Inner layer: the side of the fabric that is in contact with the skin while wearing. The relative error of all counts is less 3%.

**Table 3 materials-15-02647-t003:** The calculated values of the conventional diameters of spun yarns used for production of the knitted fabrics.

Brand Name of Yarn	Fiber Content, %	Density, kg/m^3^		Conventional Diameter d, µm
		Of Individual Fiber [19,20]	Of Yarn	
METAFINE.X.95^®^	95% meta-aramid	1.46 × 103	1.459 × 10^3^	119.70
	5% para-aramid	1.44 × 103		
METALEN^®^	50% meta-aramid	1.46 × 103	1.485 × 10^3^	118.65
	50% viscose FR	1.51 × 103		

**Table 4 materials-15-02647-t004:** Grading of moisture management test (MMT) indices (AATCC 195).

Grade Index		1	2	3	4	5
Wetting time, WT, s	Top-WT_T_	≥120	20–119	5–19	3–5	<3
		No wetting	Slow	Medium	Fast	Very fast
	Bottom-WT_B_	≥120	20–119	5–19	3–5	<3
		No wetting	Slow	Medium	Fast	Very fast
Absorption rate, AR, %/s	Top-AR_T_	0–9	10–29	30–49	50–100	>100
		Very slow	Slow	Medium	Fast	Very fast
	Bottom-AR_B_	0–9	10–29	30–49	50–100	>100
		Very slow	Slow	Medium	Fast	Very fast
Max wetted radius, MWR, mm	Top-MWR_T_	0–7	8–12	13–17	18–22	>22
		No wetting	Small	Medium	Large	Very large
	Bottom-MWR_B_	0–7	8–12	13–17	18–22	>22
		No wetting	Small	Medium	Large	Very large
Spreading speed, SS, mm/s	Top-SS_T_	0–0.9	1–1.9	2–2.9	3–4	>4
		Very slow	Slow	Medium	Fast	Very fast
	Bottom-SS_B_	0–0.9	1–1.9	2–2.9	3–4	>4
		Very slow	Slow	Medium	Fast	Very fast
Accumulative one-way transport capability, AOTI, %		<−50	−50 to 99	100–199	200–400	>400
		Poor	Fair	Good	Very good	Excellent
Overall moisture management capacity, OMMC		0–0.19	0.2–0.39	0.4–0.59	0.6–0.8	>0.8
		Poor	Fair	Good	Very good	Excellent

**Table 5 materials-15-02647-t005:** Grading of moisture management test (MMT) indices (AATCC 195).

Property	Measuring Unit	Level		
		A	B	C
**Warm climate**				
Liquid moisture management: overall moisture management capacity (OMMC) [AATCC 195]	index	index ≥ 4	4 > index ≥ 3	index ˂ 3
	1–5	(OMMC ≥ 0.6)	(0.6 > OMMC ≥ 0.4)	(OMMC ˂ 0.4)
Thermal resistance, R_ct_ [EN ISO 11092]	m^2^ K/W	R_ct_ ≤ 0.015	0.015 ˂ R_ct_ ≤ 0.03	0.03 ˂ R_ct_ ≤ 0.04
**Cold climate**				
Liquid moisture management: overall moisture management capacity (OMMC) [AATCC 195]	index	index ≥ 4	4 > index ≥ 3	index ˂ 3
	1–5	(OMMC ≥ 0.6)	(0.6 > OMMC ≥ 0.4)	(OMMC ˂ 0.4)
Thermal resistance, R_ct_ [EN ISO 11092]	m^2^ K/W	R_ct_ ≥ 0.08	0.08 > R_ct_ ≥ 0.05	R_ct_ ˂ 0.05

**Table 6 materials-15-02647-t006:** Summary data of moisture management properties of fabrics.

Code of Fabric	WT_T_, s (Grade)	WT_B_, s (Grade)	AR_T_, %/s (Grade)	AR_B_, %/s (Grade)	MWR_T_, mm (Grade)	MWR_B_, mm (Grade)	SS_T_, mm/s (Grade)	SS_B_, mm/s (Grade)	AOTI, %; (Grade)	OMMC; (Grade)
1A	6.37 ± 1.56 (3)	2.75 ± 0.33 (5)	24.56 ± 3.42 (2)	38.84 ± 0.99 (3)	11 ± 2.24 (2)	23 ± 2.74 (5)	1.58 ± 0.34 (5)	4.37 ± 0.21 (5)	325.11 ± 23.15 (4)	0.75 ± 0.03 (4)
2A	7.38 ± 0.39 (3)	3.09 ± 0.24 (4)	20.95 ± 1.80 (2)	40.58 ± 1.21 (3)	11 ± 2.24 (2)	20 ± 0 (4)	1.36 ± 0.16 (5)	3.84 ± 0.21 (4)	335.77 ± 14.04 (4)	0.75 ± 0.03 (4)
3A	6.93 ± 0.43 (3)	3.88 ± 2.50 (4)	32.54 ± 2.06 (3)	45.87 ± 7.48 (3)	10 ± 0 (2)	21 ± 2.24 (4)	1.41 ± 0.18 (5)	4.26 ± 0.20 (5)	353.68 ± 10.77 (4)	0.80 ± 0.03 (4)
4A	5.19 ± 1.64 (3)	2.92 ± 0.10 (5)	22.29 ± 2.72 (2)	42.54 ± 0.58 (3)	12 ± 2.74 (2)	21 ± 2.24 (4)	1.86 ± 0.54 (5)	4.01 ± 0.16 (5)	331.19 ± 32.112 (4)	0.76 ± 0.04 (4)
5A	3.29 ± 0.20 (4)	2.77 ± 0.31 (5)	30.96 ± 1.83 (3)	39.72 ± 0.66 (3)	19 ± 2.24 (4)	22 ± 2.74 (4)	3.45 ± 0.31 (4)	4.22 ± 0.37 (5)	178.02 ± 28.75 (3)	0.59 ± 0.03 (3)
6A	3.56 ± 0.26 (4)	2.73 ± 0.31 (5)	32.40 ± 1.08 (3)	42.17 ± 0.65 (3)	18 ± 2.74 (4)	22 ± 2.74 (4)	3.40 ± 0.36 (4)	4.46 ± 0.31 (5)	197.31 ± 15.77 (3)	0.61 ± 0.02 (4)
7A	3.74 ± 0.19 (4)	2.94 ± 0.18 (5)	32.23 ± 1.17 (3)	42.41 ± 0.86 (3)	16 ± 2.24 (3)	23 ± 2.74 (5)	2.96 ± 0.24 (4)	4.56 ± 0.32 (5)	205.15 ± 8.87 (4)	0.62 ± 0.01 (4)
8A	3.67 ± 0.51 (4)	2.9202 ± 0.58 (5)	33.937 ± 0.92 (3)	42.9662 ± 1.18 (3)	18 ± 2.74 (4)	23 ± 2.74 (5)	3.4970.37 (4)	4.61 ± 0.27 (5)	197.84 ± 10.50 (3)	0.62 ± 0.01 (4)
1B	13.67 ± 0.59 (3)	3.16 ± 0.69 (4)	9.05 ± 3.74 (1)	52.19 ± 1.98 (4)	8 ± 2.74 (2)	25 ± 0 (5)	0.6553 ± 0.46 (1)	4.30 ± 0.34 (5)	641.10 ± 31.14 (5)	0.86 ± 0.01 (5)
2B	8.63 ± 2.76 (3)	3.03 ± 0.71 (4)	12.81 ± 5.00 (2)	55.72 ± 2.1 (4)	10 ± 0 (2)	25 ± 0 (5)	1.11 ± 0.22 (2)	4.71 ± 0.56 (5)	596.93 ± 38.78 (5)	0.88 ± 0.01 (5)
3B	11.14 ± 1.43 (3)	2.92 ± 0.44 (5)	8.92 ± 2.04 (1)	54.50 ± 1.76 (4)	10 ± 0 (2)	25 ± 0 (5)	0.85 ± 0.11 (1)	4.75 ± 0.28 (5)	662.28 ± 42.37 (5)	0.87 ± 0.01 (5)
4B	11.61 ± 0.46 (3)	3.46 ± 0.16 (4)	14.30 ± 1.43 (2)	57.48 ± 0.72 (4)	8.33 ± 2.88 (2)	25 ± 0 (5)	0.756 ± 0.16 (1)	4.13 ± 0.15 (5)	751.92 ± 22.72 (5)	0.88 ± 0.002 (5)
5B	6.77 ± 2.13 (3)	5.85 ± 1.94 (3)	38.27 ± 2.98 (3)	52.46 ± 8.48 (4)	17.50 ± 2.74 (3)	20 ± 0 (4)	2.54 ± 0.52 (3)	2.96 ± 0.49 (3)	192.92 ± 27.00 (3)	0.59 ± 0.05 (3)
6B	3.99 ± 0.48 (4)	3.44 ± 0.18 (4)	33.83 ± 2.65 (3)	54.55 ± 1.27 (4)	17 ± 2.74 (3)	20 ± 0 (4)	3.05 ± 0.17 (4)	3.56 ± 0.13 (4)	263.23 ± 39.37 (4)	0.69 ± 0.04 (4)
7B	3.67 ± 0.21 (4)	3.30 ± 0.28 (4)	33.61 ± 2.26 (3)	53.84 ± 1.14 (4)	19 ± 2.24 (4)	23 ± 2.74 (5)	3.25 ± 0.17 (4)	3.91 ± 0.24 (4)	276.67 ± 36.44 (4)	0.72 ± 0.05 (4)
8B	3.87 ± 0.11 (4)	3.46 ± 0.32 (4)	34.15 ± 1.12 (3)	54.12 ± 1.21 (4)	20 ± 0 (4)	21.67 ± 2.88 (4)	3.42 ± 0.14 (4)	3.83 ± 0.16 (4)	276.06 ± 13.26 (4)	0.72 ± 0.02 (4)

Note: The letter “T” next to the indices means the upper sensor’s data, i.e., the parameters of the inner layer of the fabrics, and the letter “B” means the bottom sensor’s data, i.e., the parameters of the outer layer of the fabrics.

**Table 7 materials-15-02647-t007:** Results of the parameters representing thermoregulatory properties.

Code of Fabrics Group	Water Vapor Permeability WVP, g/m^2^ 24 h ± Standard Deviation	Thermal Resistance, Rct, m^2^ K/W ± Standard Deviation	Air Permeability, mm/s ± Standard Deviation	Tightness Factor TF, tex^1/2^/mm ± Standard Deviation
1A–4A	1404 ± 37	0.063 ± 0.010	1764.1 ± 26.6	1.46 ± 0.06
5A–8A	1624 ± 27	0.068 ± 0.011	1861.4 ± 49.7	1.71 ± 0.02
1B–4B	1612 ± 70	0.082 ± 0.013	1691.4 ± 71.1	1.42 ± 0.06
5B–8B	1349 ± 39	0.052 ± 0.00	1812.7 ± 82.7	1.64 ± 0.04

**Table 8 materials-15-02647-t008:** Evaluation of charred area parameters after flame application to the fabrics.

Code of Fabric	Charred Area Parameters							
	Specimens in Longitudinal Direction				Specimens in Cross Direction			
	Area, mm^2^	Perimeter, mm	Length, mm	Width, mm	Area, mm^2^	Perimeter, mm	Length, mm	Width, mm
1A	4791	1030	40.86	25.52	5054	1102	45.55	25.47
2A	5273	1090	42.27	29.08	4108	976	37.45	24.94
3A	7196	1154	51.74	31.40	5397	1148	47.07	27.21
4A	6324	1182	46.91	30.22	3621	956	30.22	25.50
5A	5674	1148	47.26	27.06	3736	1090	68.66	16.11
6A	5479	1156	42.24	27.56	4835	1170	37.43	31.23
7A	4162	1086	41.68	22.20	5939	1114	31.07	32.62
8A	5588	1190	49.09	27.03	4629	1058	36.15	29.32
1B	5254	1124	44.88	26.19	5869	1148	47.24	27.04
2B	5473	1138	40.34	31.61	3660	1010	40.18	21.90
3B	5274	1096	43.39	28.57	2794	1188	38.43	15.59
4B	5317	1090	47.87	24.96	5088	1092	46.37	27.21
5B	3930	970	39.84	21.40	5428	1034	43.77	27.51
6B	3637	946	33.98	20.44	4758	1080	45.35	24.86
7B	7586	1238	48.56	35.61	5479	1102	45.38	28.40
8B	4476	1056	41.53	25.32	5723	1096	45.85	31.61

## Data Availability

Not applicable.

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
