# Peer review of "Flammability and Thermoregulation Properties of Knitted Fabrics as a Potential Candidate for Protective Undergarments"

_materials, 2022, doi:10.3390/ma15072647_

Round 1

Reviewer 1 Report

The paper is devoted to the investigation of the properties of the knitted fabrics used for the protective non-flammable underwear. The authors have tested the fabric prepared of the commercially available fibers. To the reviewer's mind, more attention in the paper should be paid to the properties of the fiber, not only to the fabric itself.

The text also contains some misprints, and should be proofreaded.

Reviewer 2 Report

I would like to congratulate the authors for the great quality of the paper in order to investigate the flammability and thermal comfort properties of two-layer knitted fabrics, two groups of FR aramid fiber fabrics.

To further improve this great work, I'm reporting some topics to be modified.

- On line 37, the symbol “÷” is displayed, which is normally used in mathematical expressions. With that, I suggest changing this symbol.

- On line 44, the error in the symbol “°” C appears, which must be corrected.

- The text must be completely proofread to look for these small typos.

- Figure 1 presents images with little clarity.

- Table 2 presents wrong symbols “®”.

- Line 169 has double space.

- Table 5 has the wrong formatting.

- The paragraph of line 179 is with wrong line spacing.

- Is there a specific standard or reference on the water vapour permeability?

- Figure 3 presents its caption incorrectly “Tl”.

- Is there no reference more current than the year 2018?

Reviewer 3 Report

Please find the enclosed file.

Round 2

Reviewer 3 Report

Nil